# Perspectives on the Use of Algae in Agriculture and Animal Production

**Joël Fleurence**

Sea, Molecules and Health Laboratory EA 21060, University of Nantes, Houssiniere Path, CEDEX 3, 44322 Nantes, France; joel.fleurence@univ-nantes.fr; Tel.: +33-251-125-660

**Abstract:** Algae have been used in agriculture as fertilizers for a long time. Recently, they have also been applied to crops as biostimulants that target plant growth promotion and tolerance to biotic (herbivores, fungi, bacteria, viruses) or abiotic stresses. In addition, algae contain bioactive compounds that have been shown to maintain the health of domestic animals or aquaculture species. This opinion piece highlights different aspects of the present use of algae in agriculture and animal production and their future perspectives.

**Keywords:** algae; agriculture; animal feeding; plant and animal health

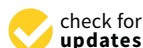



The application of seaweeds for soil fertilization is an ancestral practice. The Romans Columella and Palladius suggested the use of algal manure for the cultivation of cabbage or fruit trees such as pomegranates or citrus trees [1,2]. In Ireland, the coastal populations frequently used the "goémon" (*Ascophyllum* sp., *Fucus* sp.) to prepare the soil before the cultivation of potatoes [3]. In France, especially in Britanny, the stranded seaweeds were spread on the fields to prepare the plantation of cauliflower, artichokes, potatoes or fruit trees [4]. Algae have also been used to feed farm animals. In Norway, brown seaweeds were frequently fed to livestock after desalting and boiling. The main species considered by this process are *Alaria esculenta*, known under the Norwich name "Kutara" (cow algae), and *Ascophyllum nodosum*, called "Grisetang" (pig algae), respectively [3]. During the First World War, in Germany, seaweeds were included in the diet of pigs, cows, ducks and sheep. In the same period, the French army carried out some experiments on feeding horses with brown algae belonging to the genus *Laminaria*.

Nowadays, seaweeds are used in agriculture either for soil preparation (i.e., soil amendment by adding Maërl) or as biostimulants. Biostimulants are considered to be different from fertilizers and they induce plant growth when used in very small quantities [2]. The seaweed extracts involved to this type of activity are mainly obtained from brown seaweed belonging to the Laminariales (*Laminaria*) or Fucales (i.e., *Ascophyllum nodosum*, *Fucus* sp.) orders (Table 1). These products are applied to crops by seed soaking, foliar spraying or direct spraying on the soil. They mainly promote seed germination and the growth of plants such as bean, cucumber, cauliflower or eggplant [5]. The presence of phytohormones such as cytokinins or abscisic acid in many seaweed extracts (Table 1) (Figure 1) probably explains the observed effects on plant growth.

With the exception of plant growth, algae extracts can also stimulate other interesting properties for agriculture. Certain molecules present in seaweed extracts promote the resistance of plants to biotic or abiotic stresses (Table 2). For example, the Laminarin polysaccharide (Figure 2) present in *Ascophyllum nodosum* shows an inducing activity of the natural defense mechanisms of plants against fungal, bacterial or viral infection [2].

**Table 1.** Examples of some products obtained from brown algae (Ochrophyta) used as stimulants of plant growth [1,6,7].

| Seaweed | Products | Molecules | Company | Country |
|---|---|---|---|---|
| *Ecklonia maxima* | Kelpak | Abscisic acid, gibberellins, cytokinins | Kelp Products | South Africa |
| *Durvillaea potatorum* | Seasol | Cytokinins (i.e., Zeatin) | Seasol International | Australia |
| *Durvillaea antartica* | Profert | - | BASF | Chile |
| *Ascophyllum nodosum* | Goëmar | Abscisic acid, auxins, cytokinins | Goëmar laboratories | France |
| | Gofar | | Gofar Agro | China |
| *Laminaria digitata* | Agrocean | Abscisic acid, auxins | Agrocean | France |
| *Fucus* sp. | Seaweed flakes | Gibberellins, auxins | Aquatic Chemical | India |

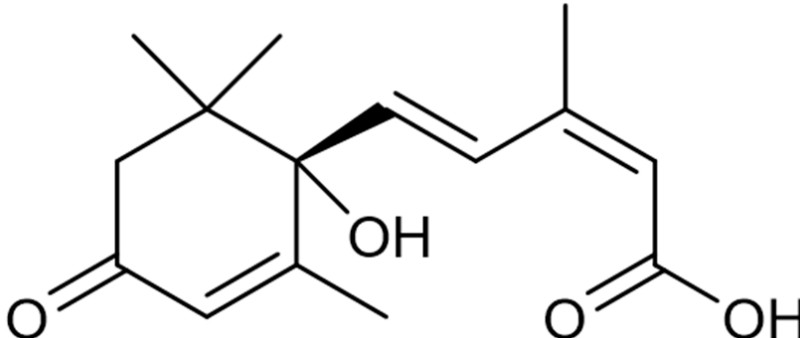

**Figure 1.** Structure of typical phytohormone (Abscisic acid) contained in *Ascophyllum nodosum*.

**Table 2.** Examples of polysaccharides contained in some brown seaweed species with an inducing activity for biochemical mechanisms involved in resistance of plants against pathogens [7].

| Seaweed Molecule Used as Elicitor | Plant | Natural Defense Mechanisms |
|---|---|---|
| Alginate | Wheat | Stimulation of phenylpropanoid pathway (Phenyl ammonia lyase (PAL) induction) |
| Alginate | Tobacco | Increased PAL activity and activation of hypersensitivity reaction to Tobacco Mosaic Virus (TMV) |
| Alginate | Japanase horseradish | Activation of Chitinase |
| Laminarin | Tobacco | Increased of pathogenesis proteins (PR) and PAL induction |
| Laminarin | Grapevine | Increased PR proteins and phytoalexin production |
| Laminarin | Rice | Induction of enzymes such as PAL and Chitinase |
| Fucoidan | Tobacco | Increased PR proteins |

Concerning abiotic stresses caused by, for example, water loss, heat or salinity, some algal extracts show positive activities which improve plant tolerance under stress situations. The brown seaweeds contain chemical compounds which increase drought tolerance. These molecules belong to the family of betaines (Figure 3) [7]. They behave as cytoplasmic protectors for cells when plants are subjected to an osmotic shock during a drought period [2,7].

The effects of the application of *Ascophyllum nodosum* extract on the tolerance of plants to salinity stress have often been described in the literature [2,7]. The molecular mechanism

responsible for this tolerance is only partially known. Betaines also seem to be involved in the resistance of salinity [1]. Likewise, treatment with an extract of *A. nodosum* significantly improves the frost tolerance in barley during winter. As above, the mechanism of this tolerance is not fully known.

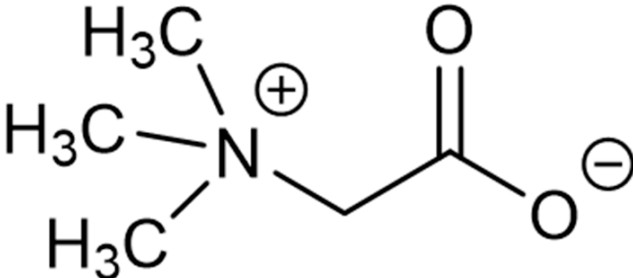

**Figure 2.** Structure of Laminarin extracted from *Ascophyllum nodosum*.

**Figure 3.** Structure of betaine contained in *Ascophyllum nodosum* extract.

The use of seaweed in animal feed has remained a traditional practice in Europe, for sheep, pigs, horses, and cattle [8].

Nowadays, there are some examples of including seaweeds in the diet of laying hens or fish produced by aquaculture. The addition of *Ulva* meal (i.e., 5% of the food ration) into the diet of laying hens was carried out in France to improve the yellow color of eggs.

Some marine animals produced by aquaculture can be fed with the addition of seaweeds to their diet. This is the case of certain mollusks such as abalone [9] or fish [10].

For fish, the addition of *Ulva* meal or wet feed including alginate improves the immune status of species such as sea bream or Atlantic salmon [9]. For abalone, the consumption of red seaweed such as *Palmaria palmata* could be useful to protect this mollusk against the pathogen *Vibrio harveyi*. Indeed, the extract of *P. palmata* obtained after organic solvent extraction shows a significant antibacterial activity against the abalone pathogen [9].

Algae and more particularly microalgae have interesting properties for animal health. The addition of the cyanobacteria *Spirulina* sp. in the chicken diet showed a positive effect on the animal immune system [11]. Some molecules such as polysaccharides, phycobiliproteins, or polypeptides have been described for their antibacterial, antiviral and antitumoral activities. These compounds are present in most microalgae used for animal feeding. The species concerned belong to genus *Chlorella* or the red algae *Porphyridium cruentum*.

**Conclusions**

Seaweeds, microalgae and cyanobacteria are curently used in agriculture and animal production. The more precise characterization of the molecules involved, in particular in plant and animal health, is a challenge for the years to come. Progress in research

in this area should be useful and necessary for the establishment of new methods for algal resource valorization. Recently, a study showed that the addition of the red seaweed *Asparogopsis taxiformis* to the diet of young ruminants reduced enteric methane production by over 80% [12]. This result reinforces the interest of using algae in the future to limit methane emissions generated by animal production.

The input of this marine biomass into agriculture and animal production sectors should promote a new agricultural concept that is less dependent of the chemical products frequently used.

**Funding:** This research received no external funding.

**Institutional Review Board Statement:** Not applicable.

**Informed Consent Statement:** Not applicable.

**Conflicts of Interest:** The author declares no conflict of interest.

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
