# Peer review of "Perspectives on the Use of Algae in Agriculture and Animal Production"

_phycology, doi:10.3390/phycology1020006_

Round 1

Reviewer 1 Report

Manuscript ID: phycology-1416015

Dear Editor and Author,

The manuscript entitled “What Future for Using of Algae in Agriculture and in Animal Production?” is a well-written short review aiming to address different aspects of the use of algae in agriculture and animal production. However, I would expect the author to provide a more personal perspective on the actual challenges/limitations/advantages of algae in agriculture and animal production.

In the Introduction section, the author mentions several examples about the historical use of algae. In the Azores, for instance, several species of seaweeds were also traditionally used either as food (e.g. Fucus spiralis, Porphyra spp., Laurencia spp. and Osmundea spp.) or for extraction of chemical products (Pterocladiella capilacea and Gelidium spp.). It would be a location of interest for being included, as it has arising as a promising hotspot of biodiversity.

Line 31: Replace “Fucale” to “Fucales” and “Laminariale” to “Laminariales”.

Table 1: “Fucus sp.” Cannot be placed into the category “Algal species”, as no species is in fact detailed (only the genus Fucus). I suggest the author to replace the column title to simply “Algae” (or seaweed). Plus, I recommend the author to include the botanical identifier of each species/genus and to characterize if they belong to Ochrophyta, Rodophyta, or Chlorophyta divisions. Replace “Name of products” simply by “Product”.

Line 42: Replace “laminarin polysaccharide” by “polysaccharide Laminarin”.

Line 48: Correct the typo in reference 6.

Table 2: Correct “phenylpropanoid” and “fucoidan”

Line 53: Correct the typo in reference 6.

Line 56: Replace “The effects of Ascophyllum nodosum extract application” by “The effects of the application of Ascophyllum nodosum extract”.

Line 65: Replace “inclusion the” by “including”.

Lines 66-68: Please, add a reference to support this sentence. Plus, I am not sure what “eg” means. Is it an example? If so, you should replace it by “e.g.,”.

Line 74: Replace “Indeeds” by “Indeed”.

Line 84: Replace “ready” by “already”.

Line 89: Replace “should be” by “should”.

Author Response

Thank you for your positive suggestions. I included in the revised manuscript the changes proposed by you.

Pr Joël Fleurence

Reviewer 2 Report

Line 2: delete “in”

Line 20: delete “The” Algae

Line 21: Main species concerned to this process were Alaria esculenta and Ascophyllum nodosum, known under the Norwich names as “Kutara” (cow algae) and “Grisetang” (pig algae) respectively.

Line 24: delete the first “In Germany”; During the First…in Germany,…

Line 25: sheeps.

Line 26: …to the genus Laminaria.

Line 27: (i.e.

Line 31: Laminariales – Fucales (i.e.

Reviewer comment to sentence in lines 34-36: please, could you include a reference justifying this affirmation?

Reviewer comment: please in the final edition, could you try to have the table 1 as a whole in a page? A broken table is not well edited. Also include a line under the first row: Algal species – Commercial product – Company – Country

Reviewer comment: please in the final edition, could you try to have the table 2 as a whole in a page? A broken table is not well edited. Also include a line under the first row: Seaweed molecule…

Line 49: change … salinity stresses by ...salinity induced,…

Line 50: extracts show positive activities which improve plant tolerance under stress situations.

Lines 51-52: please include a reference justifying this.

Lines 56-57: please include references justifying this.

Line 63: sheeps

Line 63: Seaweed resources were…

Line 66: (i.e. 5%...

Line 70-71: For fishes,…

Line 75:…shows…

Line 77: chicken diet show a positive…

Line 79: antibacterial, antiviral and antitumoral activities.

Line 81: belong to the genus Chorella or the red algae Porphyridium cruentum.

Line 84: Seaweeds, microalgae and cyanobacteria are ready…

Line 89: should promote a new agricultural concept…

Author Response

Thank you for your advices and positive suggestions. I included all changes suggested by you.

Many thanks

Pr Joël Fleurence

Reviewer 3 Report

The manuscript "What Future for Using Algae in Agriculture and Animal Production?" by Joël Fleurence is undeniably a hot-topic in algae research and investigation, which is why I consider it for publication in the journal Phycology.

The work is well-written in general.

However, it lacks recent published articles on this topic of research, as well as commercially available products, for a review.

The language is quite generic, with no explanation of how algae (directly, their extracts or isolated chemicals) can benefit plant health and animal feed.

It also lacks the presentation of case studies and reported results.

I've attached my comments on the manuscript.

Author Response

It is a mini-review as requested by Phycology.

1) About the lacks of recents publications on the topic, my reply is the following:

 Line 93 [2] In press (2022)

Lines 98, 102, 103 , 104, 110 : [5] 2015, [7] (2016), [8] 2015 [10] (2021)

2) It lacks commercially available products for a review ???

Please see Table 1 with commercially products cited (eg Seasol, Profert, Goëmar, Gofar, Agrocean)

3) No explanation how algae can benefit plant health or animal feed ?????

Figures 1, 2, 3: seaweed molecules involved to the improvement of growth and plant resistance to biotic and abiotic stresses!

Table 2: Description of seaweed molecules with an inducing activity of defense mechanism of plant against pathogens.....!!!

Please read the text associated with the  figures (Lines 40 to 61).

For the benefices to animal feed:

The effects to the use of algae purchased as a meal , as organic extracts or molecules are given in Lines 65 to 80.

4) It lacks the presentation of case studies?

I do not understand...what are the references? Not references to studies? What do you think for the reference 8?

General comment: It is my short reply for a short review on my mini-review

Round 2

Reviewer 3 Report

In terms of commercial products. There are several other seaweed-based products available, including Biosept 33SL, Bio-Algeen S90 Plus, Labimar 10S, Kelpak SL, Lysodin Alga-Fert, and Vaxiplant SL (see https://doi.org/10.3390/agronomy10020173 and https://doi.org/10.3390/su13084485).

If it is at all possible, t he author should include a column in table 1 describing the bioactive compounds in these products. 

Regarding the case studies, I mean that growth rates, for example, should be presented. Does it truly make a difference on agricultural productivity when seaweed compounds/extracts are used on crop plants? It would be ideal if these values were presented.

Author Response

Thank you for your suggestions. I added a column in table 1 as requested. There are a lot of seaweed products....I focused on the products more cited in the scientific publication. It is difficult to give  crop productivity values after seaweed treatment. There are different according the plants.

Pr Joël Fleurence